# Ligand-Length Modification in CsPbBr_3_ Perovskite Nanocrystals and Bilayers with PbS Quantum Dots for Improved Photodetection Performance

**DOI:** 10.3390/nano10071297

**Published:** 2020-07-02

**Authors:** Juan Navarro Arenas, Ananthakumar Soosaimanickam, Hamid Pashaei Adl, Rafael Abargues, Pablo P. Boix, Pedro J. Rodríguez-Cantó, Juan P. Martínez-Pastor

**Affiliations:** 1UMDO, Instituto de Ciencia de los Materiales, Universidad de Valencia, 46071 Valencia, Spain; ananthakumar.soosaimanickam@uv.es (A.S.); hamid.pashaeiadl@uv.es (H.P.A.); rafael.abargues@uv.es (R.A.); Pablo.P.Boix@uv.es (P.P.B.); 2Intenanomat SL, Catedrático José Beltrán 2, 46980 Paterna, Spain; p.rodriguez@intenanomat.es

**Keywords:** ligand-exchange, 3-mercaptopropionic acid, perovskite nanocrystals, photodetectors

## Abstract

Nanocrystals surface chemistry engineering offers a direct approach to tune charge carrier dynamics in nanocrystals-based photodetectors. For this purpose, we have investigated the effects of altering the surface chemistry of thin films of CsPbBr_3_ perovskite nanocrystals produced by the doctor blading technique, via solid state ligand-exchange using 3-mercaptopropionic acid (MPA). The electrical and electro-optical properties of photovoltaic and photoconductor devices were improved after the MPA ligand exchange, mainly because of a mobility increase up to 5 × 10^−3^
${\mathrm{cm}}^{2}/\mathrm{Vs}$. The same technology was developed to build a tandem photovoltaic device based on a bilayer of PbS quantum dots (QDs) and CsPbBr_3_ perovskite nanocrystals. Here, the ligand exchange was successfully carried out in a single step after the deposition of these two layers. The photodetector device showed responsivities around 40 and 20 mA/W at visible and near infrared wavelengths, respectively. This strategy can be of interest for future visible-NIR cameras, optical sensors, or receivers in photonic devices for future Internet-of-Things technology.

## 1. Introduction

All-inorganic cesium lead halide perovskites, with formulation CsPbX_3_ (*X* = Cl, Br, I), have been proposed for a large number of optoelectronic applications because of their unique properties, such as a large optical absorption cross section and high photoluminescence quantum yield (PLQY) [1]. Besides, these materials, which were engineered as nanocrystals for the first time in 2015 [2], exhibit a relatively low concentration of defects [3] and enhanced endurance to ambient environment as compared to their organic–inorganic analogues [4], while it also allows for a flexible bandgap tunability with narrow emission lines, being the ideal material for next generation of light-emitting diodes (LEDs) and display applications [5]. In the field of photodetectors, these materials can display high responsivities in the form of nanowires [6,7], because of their long carrier lifetimes and fast charge transfers [8], and monolayers (or few-layer nanosheets) that use a metal-semiconductor-metal (MSM) configuration [9,10]. The MSM architecture was also applied to the development of optoelectronic devices based on films of nanocrystals (nanocube shape) [11,12,13]. However, a few publications explore photodetector architectures based on heterojunctions, as in the case of solar cells. While these configurations are arguably more complex, they facilitate the introduction of functional interfaces, which can increase the charge carrier selectivity at the contacts, enabling potentially higher detection performance.

The use of perovskite nanocrystals (PNCs) in the form of thin films offer additional advantages as compared to the use of perovskite polycrystalline thin films, such as the composition control during the PNC synthesis [1], the self-assembling of PNCs leading to 3D superlattices or supercrystals [14], and the possibility of carrying out a post-synthetic anion or cation exchange, allowing to tune and optimize the resulting bandgap [15]. An ordered assembly of nanocrystals can enable 3D electronic coupling between them, thus enhancing charge carrier transport in the superlattice structure. This kind of structure can be obtained with by inexpensive thin film solution deposition technique such as doctor blading, which allows for a precise thickness control over large areas with a minimum waste of material as it was proven in our previous publications based on PbS quantum dots (QDs) [16,17]. Other deposition techniques, such as spin-coating, spray coating, and dip-coating, have also been employed to fabricate nanocrystal-based optoelectronic devices, some of them with relatively good figures of merit [18]. An important issue involved in the formation of compact high-quality layers of PNCs for optoelectronics originates is the insolubility of the inorganic metal halide in the processing solvents [19]. Besides, the ionic nature of perovskites restricts the use of classical polar/non-polar solvent purification strategies [20]. In this context, ligands become one of the main tools to gain control over the superlattice formation and electronic coupling between nanocrystals [21]. Particularly, when insulating long chain carboxylic acids and amine functional groups, oleic acid (OA) and oleylamine (OAm) are replaced by short chain thiols, a significant improvement was observed in the performance of photovoltaic and photocatalytic devices based on PbS QDs [21,22]. 

On the basis of previous studies on QDs, when long chain ligands are exchanged by short chain thiols, an improved carrier transport is measured because of the reduced interparticle distance and consequent efficient charge transfer between neighboring PNCs in films. Mercaptopropionic acid (MPA) is a bifunctional linker molecule that is widely used as a ligand because of its potential surface capping ability and remarkable influence on charge-transport [17,22,23,24]. In order to remove the insulating ligands from the surface of PNCs, a ligand-exchange strategy was studied in a few reports. Guopeng Li et al. used phenylethylammonium bromide (PEAmBr) in methyl acetate for replacing the conventional OAm/OA surface ligands of CsPbBr_3_ PNCs, and achieved very bright films and efficient LEDs [25]. L. Zhou et al. used MPA to couple the TiO_2_ layer and CsPbBr_3_ PNCs, resulting in an improved responsivity and detectivity [26]. PNCs are more sensitive to ligand-exchange processes using MPA than conventional IV-VI group QDs because of their ionic surface. M. Gong et al. carried out a surface-engineering process using MPA ligand of a film of CsPbI_3_ PNCs that resulted in their improved stability by reducing surface traps [27]. High photoresponsivity (˃10^6^ A/W) and high photoconductive gain (3.6 × 10^6^) was obtained with these films in graphene field-effect transistors (GFETs). Moreover, ligand exchange has been proven to be a valid strategy to improve the electrical properties of solid thin films made of nanocrystals/QDs, but also the removal of ligand excess in PNCs by washing the film with ethyl acetate solution have led to an increase of its electro-optical quality [28].

In this work, we propose the use of MPA as a ligand exchange strategy to improve the charge carrier transport in CsPbBr_3_ PNCs thin films for efficient photodiode detectors. Such films were prepared by doctor blading technique using a well purified solution of CsPbBr_3_ PNCs. In thin films deposited by layer-stacking of PNCs, the dominant transport mechanism can be assimilated to carrier hopping because of the three-dimensional electronic coupling between PNCs [29]. The Schottky-heterostructure photodevices were fabricated with/without MPA ligand exchange to study the differences in their electro-optical properties. The ligand-exchange is carried out in solid state after the deposition of the PNC thin film. MPA replaces longer OAm ligands and enhances the wave-function overlapping among nanocrystals to improve charge transport via carrier hopping mechanism. The resulting photodiodes exhibited device responsivities ~0.1 A/W and detectivities as high as 8 × 10^10^ jones (1 jones = 1 cm Hz^1/2^ W^−1^), other than open circuit voltages 0.6–0.8 V and short circuit currents of 1–3 mA/cm^2^ under solar AM1.5G illumination. These values are well above those obtained in reference samples without MPA treatment. Furthermore, we also demonstrated the compatibility and film formation of PbS QDs on top of CsPbBr_3_ PNCs, together with a ligand exchange in both layers at the same time. Our analysis showed that the thiol group (-SH) of MPA efficiently coordinates to surface Pb(II) atoms to provide a very good passivation of PNCs and QDs while reducing their interparticle spacing in the film. These findings are applied to a tandem photodetector, which exhibited a broadband responsivity levels of 40 mA/W at visible wavelengths and 20 mA/W around the telecom C-band (1525–1565 nm). 

## 2. Materials and Methods 

### 2.1. Materials

Lead(II) bromide (PbBr2, 98%, Sigma-Aldrich, Madrid, Spain), cesium carbonate (Cs2CO3, 99.9%, metals basis, Alfa Aesar, Karlsruhe, Germany), oleylamine (OAm) (technical grade, 70%, Sigma-Aldrich), oleic acid (OA) (technical grade, 90%, Sigma-Aldrich), 1-octadecene (1-ODE) (95%, Sigma-Aldrich), ethyl acetate (Sigma-Aldrich), *n*-hexane (99%, spectrophotometric grade, Sigma-Aldrich), (poly(3,4-ethylenedioxythiophene) (PEDOT:PSS), and isopropanol (99%, technical grade, Sigma-Aldrich), were used for the synthesis of CsPbBr3 PNCs and fabrication of layers. The ligand exchange was performed using 3-mercaptopropionic acid (99%, MPA, Sigma-Aldrich). All chemicals were used without further purification. 

### 2.2. Synthesis and Purification of CsPbBr_3_ Nanocrystals

Synthesis of CsPbBr_3_ PNCs was performed using the hot-injection method [2]. In detail, first Cs-oleate was prepared by mixing 0.16 g Cs_2_CO_3_, 10 mL of OA and 10 mL of 1-ODE into a 50 mL three-neck flask by heating at 120 °C in vacuum under a constant stirring. Then, the mixture was purged by N_2_ and heated at 150 °C until all of the Cs_2_CO_3_ was completely dissolved. The solution was further stored under N_2_ by keeping the temperature at 100 °C to avoid Cs-oleate precipitation. Meanwhile, 0.55 g PbBr_2_ was mixed with 20 mL of 1-ODE into a 100 mL three-neck flask. The mixture was heated at 120 °C under vacuum for 1 h, keeping a constant stirring. Then, 5 mL OA and 5 mL of OAm were separately added to the flask under N_2_ atmosphere, and rapidly heated to reach 170 °C. At this stage, 10 mL of Cs-oleate solution was injected into this mixture. Lastly, the flask was immersed into a bath ice for 5 s to stop the reaction. The solution was then centrifuged at 4700 rpm for 10 min and purified using the mixture of antisolvents, namely hexane and ethyl acetate, to achieve high purity CsPbBr_3_ PNCs. The purification was a key step to allow the isolation of the CsPbBr_3_ PNCs for further processing into thin films by the doctor blading technique and subsequent MPA ligand exchange. The CsPbBr_3_ PNCs were separated after discarding the supernatant and re-dispersed in hexane to prepare a colloidal solution with a concentration up to 50 mg/mL.

### 2.3. Characterization of PNCs

The absorption spectrum of the colloidal solution with synthesized CsPbBr_3_ PNCs in toluene exhibits a well-defined excitonic resonance at 482 nm (red curve in Figure 1a), whereas the PL spectrum is rather narrow (Full Width at Half Maximum around 25 nm) and centered at 498 nm (black curve in Figure 1a). The PLQY of CsPbBr_3_ PNCs in toluene reached a value of 55%. Transmission electron microscopy (TEM) image of the prepared CsPbBr_3_ PNCs given in Figure 1b shows that the prepared sample possess cubic morphology with the average size around 8.2 nm. The PL peak and absorbance resonance wavelengths are consistent with this average size, according to published results of CsPbBr_3_ PNCs similarly synthesized by the hot injection method [2].

### 2.4. Device Fabrication

For the purpose of obtaining a clear picture of the postulated MPA ligand exchange effects on the CsPbBr_3_ PNCs, three types of opto-electronic devices were fabricated and characterized: Schottky heterostructures (see its schematic representation in Figure 1c), interdigitated photoconductor structures, and OFET (organic field effect transistor) chips. With the analysis of these devices we can obtain different optoelectrical parameters such as mobility, electrical conductivity, and the figures of merit of photodetectors.

For the fabrication of the Schottky heterostructure, a prepatterned ITO substrate (Ossila, Sheffield, UK), a 8-Pixel photovoltaic substrate, was sonicated for 5 min in Hellmanex III (1%) (Ossila) at 70 °C, then rinsed in water/isopropanol and finally exposed to UVO (UV-Ozone) treatment for 20 min prior to the deposition of a thin layer (50 nm) of PEDOT: PSS (Sigma-Aldrich). The PEDOT: PSS was spin-coated at a speed of 3000 rpm for 40 s. Then, a cleansed 200 micrometer-filtered CsPbBr_3_ PNCs solution (free of aggregates laying at the bottom of the flask) with an optimal nanocrystal concentration of 30 to 50 mg/mL was casted on top of the PEDOT: PSS film using a commercial doctor blade applicator (model 4340, Elcometer, Manchester, UK). Subsequent layers of CsPbBr_3_ PNCs layer were deposited with a blade velocity of 1.5 cm/s with an intermediate annealing step of 2 min at 100 °C. This process was repeated several times, until a thickness of 350–400 nm was achieved. The thickness was measured by using a stylus profilometer (model Dektak 150, Veeco, Plainview, NY, USA). Finally, the film was annealed at 100 °C for 1 h under vacuum in order to remove the presence of hexane in between the nanocrystals and/or trapped by the ligands’ network. This annealing step prevents the NC layer to detach from the substrate during the ligand exchange procedure. For the ligand exchange, the film was dipped during 60 s into a solution of 10% (in volume) MPA in ethyl acetate and then rinsed with ethyl acetate to remove the MPA excess. After the treatment, the layer aspect appeared as a hazy white surface, but after drying with compressed air and annealing at 100 °C under vacuum for 1 h the layer presented a saturated green color. 

Finally, a 20-nm thick MoO_3_ (molybdenum trioxide) and a 100-nm thick gold layers were deposited by thermal evaporation under high vacuum. The shadow mask used during this thermal process was specially designed to work with these substrates (Ossila, Sheffield, UK). This multi-electrode mask defines up to eight separated devices whose individual active area is 4 mm^2^. Here, the MoO3 interlayer between the thin film of CsPbBr3 PNCs and the Au contact would act as an electron blocking layer.

Tandem photodevices integrating two stacked films of CsPbBr_3_ PNCs and PbS QDs were fabricated by using similar procedures described above. PbS QDs were synthetized according to a procedure described in our previous publications [16,29] in which a Schottky heterostructure was built by Doctor Blade to operate at telecom wavelengths (see emission and absorption spectra in Appendix A). 

The OFET test chips were purchased from Ossila, whereas metal–semiconductor–metal (MSM) interdigitated photoconductors were previously fabricated by UV photolithography and lift-off processing (the gold pads were 100 nm thick and consisted of 10 pairs of fingers spaced 20 μm each). CsPbBr_3_ PNCs were deposited on both OFET and interdigitated chips with the same doctor blade technique as the one used for the Schottky heterostructures.

### 2.5. Characterization of Films and Devices

The PLQY analysis of all samples was carried out by using an integrating sphere (Hamamatsu C9920-02 absolute PL quantum yield measurement system). Transmission electron microscopy (TEM) images were taken using JEOL 1010 microscope (JEOL, Tokyo, Japan) at the operating voltage of 100 kV. The electro-optical properties of the photodetectors and photoconductors were measured by a homemade setup based on a halogen lamp (25 mW/cm^2^ of white light intensity) focused onto a multimode optical fiber (800 µm of core diameter) attached to the entry of a DNS-300 monochromator (DeltaNu, Laramie, WY, USA), whose output was modulated at 1 kHz by means of a mechanical chopper and focused onto the fabricated photodetector by a 10x objective. We used a grating with 1200 (600) grooves per mm and blaze at 500 (1200 nm) nm for visible (near-infrared) wavelengths. For example, if the output slit of the monochromator was opened to 1 mm, a power of 18 nW was measured at a wavelength of 500 nm. The electrical signal of the photodetector was synchronously measured by a lock-in amplifier. The lamp spectrum at the monochromator’s output was measured using a calibrated Si photodetector and responsivity of tested photodetectors were determined by using the calibrated table of that Si photodetector. The current-voltage (*J*-*V*) characteristics under dark and AM1.5G illumination conditions were measured using a Keithley’s Series 2400 Source Measure Unit (SMU) and a solar simulator based on a 150 W Xe lamp (Zolix, model GLORIA-X150A) with an AM1.5G air mass filter (Newport). Light intensity was adjusted with a calibrated solar cell of silicon. The photocurrent transient times were measured with a triggered oscilloscope.

The Ossila OFET test chips were loaded for characterization on a high-density OFET test board (also from Ossila), designed to reduce leakage current, external noise, and stray capacitance. Drain-source current was measured at 1-V bias using the SMU, and the gate bias was applied using a DC power source connected to a BNC connector board. The optical properties of the layers, absorbance and photoluminescence (PL), were determined by means of a spectrograph (HR4000, Ocean Optics, Largo, FL, USA).

## 3. Results and Discussion

### 3.1. Thin Films of CsPbBr_3_ PNCs 

Thin films of CsPbBr_3_ PNCs with a thickness of 400 nm were deposited on borosilicate glass to measure PL (Figure 2a) and absorbance (Figure 2b) spectra before (red lines) and after (blue lines) the MPA ligand exchange. The PL spectra are very similar in shape, but significantly shifted to the red, from 502 to 517 nm, after the MPA treatment (see Figure 2a). A similar shift is observed for the exciton absorption resonance (Figure 2b). This redshift can be ascribed to a strong reduction in the inter-particle spacing among CsPbBr_3_ PNCs [16]. In fact, the solid-state ligand exchange of OA and OAm by MPA was confirmed by FTIR (Fourier-transform infrared spectroscopy) of the thin films. The samples were measured before and after the ligand exchange, showing a strong decrease in intensities of the aliphatic C–H stretching peaks at 2918 and 2845 cm^−1^ of methylene (–(CH_2_)*_n_*–) in long alkyl chain of OA and OAm, and the presence of a broad 3460 cm^−1^ peak of internally bonded OH stretching (from H bonding between carboxylic acid of MPA), as observed in Appendix A.

### 3.2. Electro-Optical Characterization of Schottky Heterostructures

Schottky-heterostructure samples with and without MPA-treatment were fabricated in order to explore the effects of the ligand exchange on the optoelectronic properties of the films constituted of CsPbBr_3_ PNCs. Details concerning the device architecture and the band alignment diagram can be found in Figure 1c and Appendix A.

The measured responsivity has a maximum at 500 nm of around 100 mA/W in the MPA-treated devices (blue solid curve in Figure 3a), as compared to the 7 mA/W of the pristine ones (red solid curve in Figure 3a), i.e., more than 14 times of enhancement. The external quantum efficiency (EQE) and the specific detectivity (D^*^) can be calculated following the usual expressions present in reference [30]. EQE (dotted curves in Figure 3a) and D^*^ in our MPA-treated photodiodes reach values up to 20 % and 8 × 10^10^ jones (1 jones = 1 cm Hz^1/2^ W^−1^), respectively, as compared to 1% and 4 × 10^9^ jones in the pristine ones. 

A typical photocurrent transient curve for the MPA treated photodetectors is presented in Figure 3b. The temporal response of the photocurrent was evaluated by using a white light halogen lamp source with an intensity around 25 mW/cm^2^ and chopped at 50 Hz. Rise and decay time constants of 2.0 and 1.5 ms are estimated from the time intervals required for photocurrent to reach 90% or to decay until 10% of the photocurrent peak value, respectively. These photocurrent time constants are comparable to other values found in recently reported photodevices (see Appendix A) [31,32]. The relatively long response times are attributed to charge trapping and detrapping processes that originate from relatively shallow defect levels at the surface of the PNCs, as demonstrated and discussed in literature [33,34].

In order to gain insight on the physical magnitudes determining charge transport and charge separation in the Schottky-heterostructure devices, we focus on their *J*-*V* curves and the observed changes after ligand exchange. The *J*-*V* curves provide a reliable benchmark for the evaluation of charge transport properties through the many parameters that can be derived from the fitting of the data. Specifically, the series resistance considers directly the carrier mobility and conductivity, the shunt resistance accounts for the recombination losses and the saturation current density includes carrier generation-recombination mechanism inside the device. Since the ligand exchange directly affects the performance of the hopping transport within the PNC film, these parameters are expected to shed light on the effects introduced by the MPA ligand exchange.

Despite the device stack configuration is not the optimal one for photovoltaic conversion, their respective parameters can be used to extract useful information. Figure 4a,b shows the *J*-*V* curves for both MPA-treated and pristine best-devices (averaged values listed in Table 1) where it can be seen a noticeable improvement of the photodiode properties due to the ligand exchange. The short-circuit current density *J*_sc_ is increased from 7 × 10^−3^ to 2.9 mA cm^−2^, more than 400-fold. The open-circuit voltage was also increased from 0.2 to 0.7 V. The relatively low values of *V_oc_* as compared to the effective bandgap of the CsPbBr_3_ PNCs, 2.25 eV [35], can be attributed to the lack of engineering in the ETL (electron transport layer)/perovskite and perovskite/HTL(hole transport layer) interfaces, which become significant sources of non-radiative recombination. In any case, measured values of *J_sc_* and *V_oc_* for devices based on MPA-capped CsPbBr_3_ PNCs are comparable with other published results. For instance, Hoffman et al. [36] reported *V_oc_* = 1.2 V and *J_sc_* = 4 mA cm^−2^ in solar cells 300 nm based on a 300 nm thick layer-by-layer stack of PNCs, whereas Akkerman et al. [37] reported *V_oc_* = 1.5 V and a *J_sc_* = 5 mA cm^−2^ in a similar device. Similarly, Yang et al. [38] studied a similar photodevice as a photodetector reporting *V*_oc_ = 1.2–1.3 V and *J_sc_* of 1–4 mA cm^−2^. It should be noted that the MPA ligand may result in a different electronic work function, as demonstrated for PbS QD solids, in which the work function can vary several hundreds of meVs depending on the ligand post-treatment applied [17,39].

The real diode equivalent circuit limited by the presence of the internal series resistance (*R*_s_) and the leakage current governed by the shunt parallel resistance (*R_sh_*) would be:(1)$$J=J_{sc}-J_{0}\left(e^{\frac{q\left(V+JR_{s}\right)}{mKT}}\right)-\frac{V+JR_{s}}{R_{sh}}$$
where the parameter *m* is the ideality factor, *J*_sc_ is the short-circuit current density and $J_{0}$ is the reverse saturation current density. The latter represents the electron thermionic emission, which is related to the recombination rate of the diode and its open-circuit voltage (*V_oc_*). The fitting of the experimental data to Equation (1) is not straightforward, because photodiodes based on QDs often show linear or superlinear instead of saturation-type behavior, whose main feature is related to large ideality factors (*m* > 2) [33,40]. That is, the carrier recombination mechanism within such regions needs descriptions that go beyond the Shockley–Read–Hall (SRH) model approximation. While the SRH recombination at deep trap states predicts *m* ≤ 2 [41], diodes based on nanostructured metal halide perovskites usually deviate from this behavior, with *m* >> 2 at relative low voltages (<1 V) [38,42]. 

The high value of the *m*-parameter may be the result of a broad energy tail of superficial trapping states present on the light-soaked PNCs [43] or simply because of the presence of leaking currents introduced by a certain random distribution of pinholes throughout the thin films. For thick samples or studies made under high illumination intensities, the effect of shallow traps of such energy tail is attenuated by state filling [44]. Thus, describing the photovoltaic device as two PN heterojunctions in series, instead of a single PN junction is a more heuristic approach to explain a *J*-*V* curve with a large *m*-parameter.

When the PNC layer is sandwiched between transport layers, i.e., ETL and HTL, individual semiconductor-semiconductor interfaces can be considered in both sides. Thus, the adoption of this model accounts for the formation of two active junctions [45,46]. The equivalent electrical circuit of the double heterojunction model [45] is used here to obtain the photodiode parameters of the MPA-treated and pristine photodiodes:(2)$$J=J_{sc}-J_{0}\left(e^{\frac{q\left(V+JR_{s}\right)}{(m_{1}+m_{2})KT}}\right)-\frac{V+JR_{s}}{R_{sh}}$$
(3)$$\frac{V_{1}}{m_{1}}=\frac{V_{2}}{m_{2}}=\frac{V+JR_{s}}{m_{1}+m_{2}}$$
where the parameters $J_{sc}$, $J_{0}$, $R_{s},$ and $R_{sh}$ correspond to the whole structure while *V_j_* and *m_j_* (j = 1,2) correspond to a single heterojunction. These equations hold true considering the approximation $J_{01}≅J_{02}=J_{0}$. Moreover, if both heterojunctions are ruled by the SRH mechanism (*m* ≤ 2), the sum of the ideality factors $m_{1}+m_{2}$ should be as high as 4, allowing again the SRH mechanism to describe the recombination in both heterojunctions. In order to avoid the non-linear fitting of data to Equation (2), the following useful transformations can be practiced
(4)$$-\frac{dV}{dJ}=\frac{\left(m_{1}+m_{2}\right)kT}{q}\left(\frac{1+\frac{1}{R_{sh}}\frac{dV}{dJ}}{J_{sc}-J-\frac{V}{R_{sh}}}\right)+R_{s}$$
(5)$$\ln\left(J_{sc}-J-V/R_{sh}\right)=\frac{q}{\left(m_{1}+m_{2}\right)kT}\left(V+JR_{s}\right)+\ln J_{0}$$

These equations allow extracting the ideality factors $m_{1}+m_{2}$ and $R_{s}$ with a linear regression (Equation (4)), if $\left(1+\frac{1}{R_{sh}}\frac{dV}{dJ}\;/\;J_{sc}-J-\frac{V}{R_{sh}}\;\right)$ is taken as the x-variable. Equation (5) allows to easily extract current *J_0_* from the y-intercept and using the best fitting value of $m_{1}+m_{2}$ previously obtained through Equation (4). The fitting curves (black continuous lines) using Equations (4)–(5) to experimental data (colored symbols) are shown in Figure 4c,d for MPA-treated and Figure 4d,f for pristine photovoltaic devices.

The ideality factors extracted from these fittings are large for both the MPA (*m_1_ + m_2_ =* 5 ± 1) and the pristine (6 ± 2) photodiodes (the uncertainty associated to the *m* factor was estimated from the standard deviation of fitting parameters deduced from all measured photodiodes), but still compatible with the SRH recombination mechanism under the hypothesis of the double-heterojunction. The reverse saturation current densities in the best devices were 2 × 10^−1^ (MPA) and 4 × 10^−3^ mA cm^−2^ (pristine), with average values (1.5 ± 1.0) × 10^−1^ (MPA) and (2 ± 1) × 10^−3^ mA cm^-2^ (pristine), as listed in Table 1. From these relatively high values, the barrier height can be related to *J_0_* through the thermionic emission theory for the Schottky diode:(6)$$\phi_{b}=\frac{kT}{q}ln\left(\frac{A^{*}T^{2}}{J_{0}}\right)$$
where $A^{*}$
$=A\left(m^{*}/m_{0}\right)$ is the effective Richardson constant with *A =* 120 A cm^−1^ K^−2^, where an electron effective mass $m^{*}$ = 0.126 $m_{0}$ can be adopted from literature [47]. The barrier height can be estimated from Equation (6): 0.51 V for the MPA-treated and 0.58 V for the pristine devices, which are around 0.2 V lower than other similar devices in literature [38,48]. The relatively high ideality factors combined with the low values for *ϕ**_b_* suggest that there is still room for the improvement of the MPA-treated devices, especially in terms of the Schottky-heterojunction interfaces and their corresponding energy alignments.

In addition, the *J*-*V* curves under dark conditions can be used to obtain the recombination current density *J*_r_, the diffusion current density *J*_d_, and the shunt resistance *R_sh_* (it was taken as a fixed parameter in Equations (4) and (5)) with a non-linear fitting by using the following double heterojunction equation [45]:(7)$$J=\frac{V}{R_{sh}}+J_{r}\left(e^{\frac{qV}{m_{r}kT}}-1\right)+J_{d}\left(e^{\frac{qV}{m_{d}kT}}-1\right)$$

Here the ideality factor $m_{r}$ is associated to the dual recombination current model, hence *m_r_ = m_r1_ + m_r2_* ≤ 4 (*m_r1_* and *m_r2_* being the ideality factors for both heterojunctions), whereas *m_d_ = m_d1_ + m_d2_* corresponds to the dual diffusion current model and hence can be as high as 2. 

Recombination currents *J_r_*(MPA) = 4 × 10^−5^ mA/cm^2^ (Figure 4g) and *J_r_*(pristine) = 5 × 10^−8^ mA/cm^2^ (Figure 4h) (average 4 ± 1 × 10^−5^ for MPA and 5 ± 2 × 10^−8^ mA cm^−2^ for pristine, as listed in Table 1) were obtained after fitting the *J*-*V* curves under dark conditions through Equation (7). The difference between pristine and MPA-treated devices is mainly attributed to the strong difference between dark conductivities. The dark recombination current of the pristine device is similar to those with similar Schottky-heterostructures in the literature [38]. From the same fitting procedure, we obtained a diffusion current density (*J*_d_) of around 5.10^−8^ mA cm^−2^ (average (5 ± 3) × 10^−8^ mA cm^−2^) for pristine devices, a value consistent with the large turn-on diffusion voltage (*V_bias_* > 1 V), as observed in Figure 4a,b. The diffusion component in MPA-treated devices does not play any role in the fitting of the curves and, thus, can be considered negligible.

Low dark recombination current is an essential for a high responsivity figure of merit in photodetectors. In spite of the relatively high values found for dark *J_0_*, the photodiodes can still reach good responsivity response when compared with those recently reported [11,13,49]. In fact, the extracted figures of merit of our MPA-treated photovoltaic device are comparable or better than those found in other photodetectors based on thin films of CsPbBr_3_ PNCs, as listed in Appendix A, but still worse than in similar photodevices based on QDs (0.2 A/W for NIR light was obtained by our previously developed PbS QD photodetectors using doctor blade deposition [50]). In this sense, if the charge injection/extraction of the photodiode was improved, photovoltaic devices would be more suitable for solar conversion. In any case, we have demonstrated the positive and potential benefit of the MPA ligand exchange for developing conductive thin films of CsPbBr_3_ PNCs as the basis for photovoltaic detectors. In next sections of this paper additional electronic transport properties will be obtained for future optimization of MPA-treated photodevices.

### 3.3. Electro-Optical Characterization of Photoconductors

A simpler MSM structure was adopted to characterize photoconductive properties of the thin films based on CsPbBr_3_ PNCs. In this way, a 300-nm thick absorbing perovskite layer was deposited by doctor blading onto the interdigitated pads. This structure is widely adopted for perovskites [13,27,31,51], because it provides a reliable testing workbench for new strategies with solution-processed PNCs.

The *J*-*V* characteristic curves (0 to 20 V) of the MSM interdigitated photodetectors are shown in Figure 5. The curves display the behavior of the pristine and MPA-treated films of PNCs under dark conditions and white-light illumination (25 mW/cm^2^). Noticeably, MPA-treated CsPbBr_3_ PNC films (Figure 5a) display higher dark current levels (black data symbols) and larger photocurrent (green data symbols) than the pristine ones (Figure 5b). The difference can be explained in terms of photo-sensitivities, defined as $S=\left(J_{L}-J_{D}\right)/J_{L}=\Delta \sigma /\left(\sigma_{0}+\Delta \sigma \right)$. Here, the currents $J_{L}$ and $J_{D}$ are measured under illumination and dark conditions at a given voltage, whereas $\sigma_{0}\;$ and Δσ are the conductivity under dark conditions and the photoconductivity, respectively.

Dark conductivity and photoconductivity listed in Table 2 are extracted from linear fits to the corresponding experimental curves in Figure 5 in the 0-10 V region. For higher voltages the *J*-*V* curves deviates from the linear law to a quadratic behavior. This nonlinear behavior originates from the space-charge-limited current (SCLC) effect that occurs when uncompensated charge carriers are injected into the material from the MSM contacts. In this condition, the mobility (μ) can be approximately estimated in the SCLC region (J ~ V^2^) under dark conditions by using the Mott–Gurney law [52]:(8)$$\mu =\frac{8J_{D}L^{3}}{9\varepsilon \varepsilon_{0}V^{2}}$$
where L is the channel length (20 μm), ε_0_ is the permittivity of vacuum, and ε is the relative dielectric constant of CsPbBr_3_, which is in the order of ~24 [53].

As listed in Table 2, the dark conductivity of the MPA-treated films (25 μS/cm) increases by more than one order of magnitude with respect to the value in the pristine ones (1 μS/cm), which is in line with results reported above for Schottky-heterojunctions, a difference that it is mostly due to the increase in the effective carrier mobility for the ligand-exchanged film for which we estimate ~10^−3^
${\mathrm{cm}}^{2}/\mathrm{Vs}$, as compared to the value found in pristine films, which is ~10^−4^
${\mathrm{cm}}^{2}/\mathrm{Vs}$, with a certain contribution of the background carrier concentration. That is, MPA is possibly doping CsPbBr_3_ PNCs (n_0_ around 6 × 10^14^ cm^−3^) over the background carrier concentration for PNCs with original ligands (n_0_ around 3 × 10^14^ cm^−3^).

### 3.4. Characterization of the FET Devices

Although the Mott−Gurney law is commonly used to extract the charge-carrier mobility in films of metal halide perovskites, the application of Equation (8) is only valid in trap-independent space-charge limited regime [54]. For this reason, the extraction of this parameter needs to be supported with additional analysis of *J*-*V* characteristics measured in FET transistors based on films of CsPbBr_3_ PNCs. This architecture allows to extract the mobility from the field-effect model.

The FET-mobility in the linear regime was determined using the characteristic output curves of the field-effect transistor, given by the plot of drain-source current (*I*_DS_) versus gate voltage (*V*_GS_) for a given drain voltage (*V*_DS_) bias (Figure 6). The transfer curves can be divided into two regions: the linear region and the saturation region. The slope of the linear region can be used to obtain the charge-carrier mobility using the following equation:(9)$$\frac{\partial I_{DS}}{\partial V_{GS}}=\frac{W}{L}\mu C_{S}V_{DS}$$
where *L* is the channel length (10 μm), *W* is the channel width (1 mm), and *C*_S_ = 1.15 × 10^−8^ F cm^−2^ is the capacitance of the SiO_2_ insulating layer (300 nm thick) in Ossila substrates. The drain-source voltage was set at 1V. In these conditions, the FET-mobility is found to be 0.005 ${\mathrm{cm}}^{2}/\mathrm{Vs}$ for the MPA-treated CsPbBr_3_ PNCs thin films and 5 × 10^−4^
${\mathrm{cm}}^{2}/\mathrm{Vs}$ for the pristine ones. These values are a factor 5 greater than those obtained through the analysis performed in previous section on MSM photoconductors in the SCLC regime, but in both cases, we measure an increase of carrier mobility by one order of magnitude after the MPA treatment of the films of PNCs.

### 3.5. Tandem PbS-CsPbBr_3_ Photodiodes

Considering the beneficial effects of the MPA ligand-exchange on the CsPbBr_3_ PNCs thin films, we propose its single-step application to a PbS QDs and CsPbBr_3_ PNCs heterojunction. Given that PbS QDs operate at telecom wavelengths (1550 nm), their integration with CsPbBr_3_ PNCs as two stacked layers would enable a broadband photodetection over a large region of optical frequencies. This kind of heterojunction has well established precedents in literature. Photodevices based on semiconductor heterostructures are a promising strategy for overcoming the Shockley−Queisser limit, while the solution processability of colloidal nanocrystals allows to keep the production costs well very low [55]. In the particular case of metal halide perovskite solar cells, the PbS QDs can be used as co-sensitizers. The addition of PbS QDs into CH_3_NH_3_PbI_3_ (MAPbI_3_) precursors was found to be beneficial for the crystallization of perovskites [56]. Also, the processing of PbS QDs as hole transporter layers (HLT) for planar heterojunction perovskite solar cells was recently demonstrated [34]. PbS QDs can readily substitute the spiro-type HTLs, while offering an increased absorption range of the solar spectrum and/or multi-wavelength LEDs due to possible exciplex states between the perovskite and QDs [57]. In the case of photodetectors, the benefits of this hybridization were also demonstrated in another study of CsPbBr_3_ bulk perovskite layers and PbS QDs-based phototransistors with a wide response spectrum, from 400 to 2100 nm [57,58]. From the point of view of the device fabrication, both PbS QDs and CsPbBr_3_ PNCs are fully compatible since they are formulated in hexane using OAm as ligand. Moreover, both deposition and ligand exchange for QDs and PNCs follow the same procedure.

Figure 7 shows the responsivity curve of our tandem-like prototype photodiode in the visible and near-infrared spectral regions. Device architecture and band alignment schemes are displayed in the inset of Figure 7. The MPA ligand exchange was carried out to the whole structure after the deposition of both PNC and QD films with the same parameters used in the case of Schottky-heterostructure based on single layers of CsPbBr_3_ PNCs. The highest responsivity is found for visible wavelengths (green curve in Figure 7): 40 and 25 mA·W^−1^ at 430 nm and the exciton absorption resonance of CsPbBr_3_ PNCs, respectively. Furthermore, the tandem photodetector exhibits a noticeable response in the near-infrared spectral region due light absorption of PbS QDs (red curve in Figure 7), reaching a second maximum of around 20 mA·W^−1^ at 1550 nm. While in this spectral range the obtained responsivity is one order of magnitude smaller than previously reported results in single PbS QD-Schottky heterojunctions fabricated in a similar way [16], it is worth to remark the significantly thinner (≈100 nm) PbS layer used in the tandem photodiode. The present device is a proof of concept of the feasible integration of two complementary nanomaterials that must be further optimized in a number of different ways (thicknesses of the stacked layers, PNC and PbS bandgaps, photodiode architecture, …) according to the application (optical sensors, visible-NIR cameras, photoreceivers or multiwavelength LEDs for integrated photonics, solar cells, …).

## 4. Conclusions

We present a solid-state ligand exchange procedure to replace the original long-chained surface ligands (long-chain aliphatic molecules like oleic acid) of CsPbBr_3_ PNCs with much shorter MPA ligand molecules. Thin films of densely packed CsPbBr_3_ PNCs were deposited by means of this method, layer-by-layer up to an optimal thickness of 300–400 nm. As a result, Schottky heterojunction photodetectors could be achieved using doctor blade deposition technique. The resulting devices exhibited and enhanced photovoltaic detection performances. In this way, the short inter-particle distance imposed by MPA ligands led to mobilities in the order of 5 × 10^−3^
${\mathrm{cm}}^{2}/\mathrm{Vs}$, as determined by FET mobility measurements, which are one order of magnitude greater than those measured in pristine films (without ligand exchange), 5 × 10^−4^
${\mathrm{cm}}^{2}/\mathrm{Vs}$. This mobility enhancement is a key factor to obtain 20-fold higher photoconductivity, with the responsivity increasing from 7 to 100 mA/W (and a detectivity as high as 8 × 10^10^ jones) in our Schottky-based heterostructures. In order to extend the applicability of the MPA solid-state ligand exchange processing, a tandem structure was integrated in the same photovoltaic architecture: a film of PbS QDs on top of a first film made of CsPbBr_3_ PNCs. The resulting device has a wide spectral response, ranging from the UV to the NIR, offering responsivity levels of 40 mA/W at visible wavelengths and 20 mA/W around the telecom C-band (1525–1565 nm). These results can be the basis of future photodiode arrays, CMOS-like cameras operating simultaneously at visible and NIR wavelengths or low-cost applications in the fields of IOT and optical sensing.

## Figures and Tables

**Figure 1 nanomaterials-10-01297-f001:**
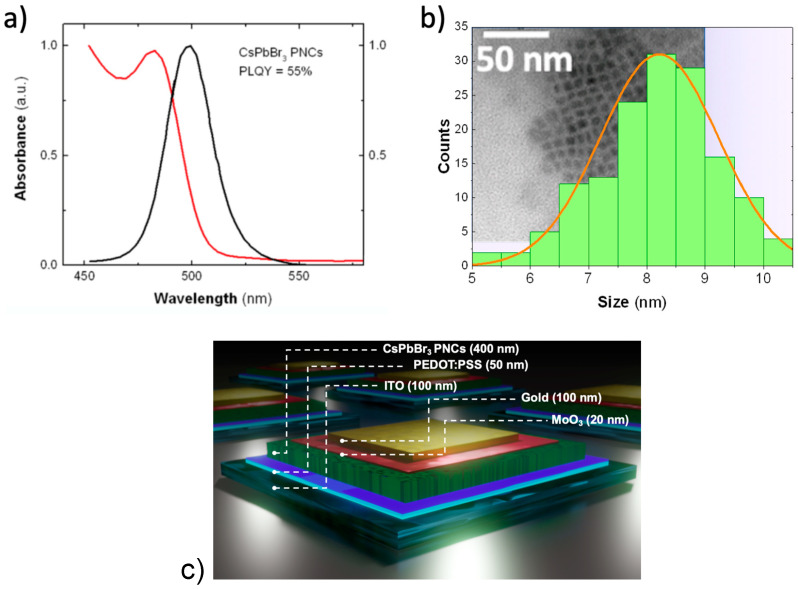
(**a**) UV-visible and PL spectrum of synthesized CsPbBr_3_ perovskite nanocrystals (PNCs); (**b**) size distribution of CsPbBr_3_ PNCs as extracted from TEM images as the one shown in the inset; (**c**) schematic representation of the device architecture.

**Figure 2 nanomaterials-10-01297-f002:**
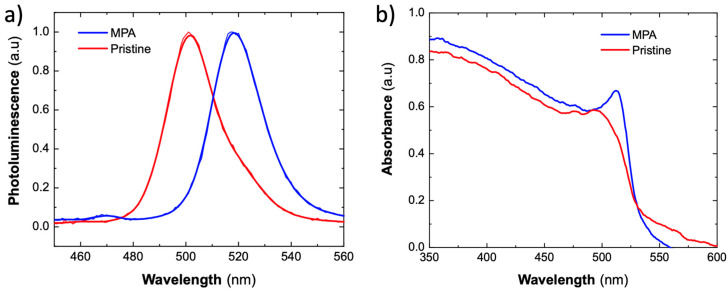
Optical characterization of the 350–400 nm CsPbBr_3_ PNCs 400 nm layers before (red curves) and after 3-mercaptopropionic acid (MPA) ligand exchange (blue curves). (**a**) Photoluminescence, measured by backscattering and collecting to a spectrograph. (**b**) UV-Vis absorbance spectra.

**Figure 3 nanomaterials-10-01297-f003:**
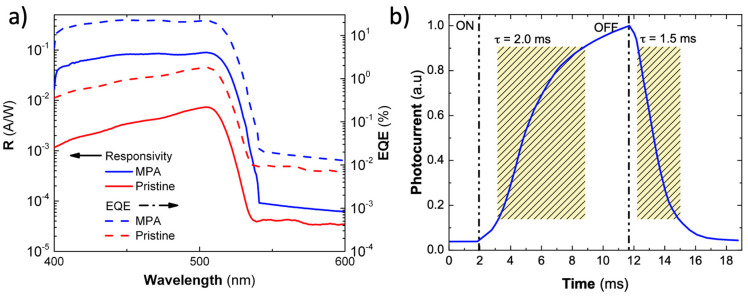
(**a**) Responsivity curve at 0 V bias of the Schottky photodiodes prepared with MPA ligand exchange (blue solid line) and pristine (red solid line) CsPbBr_3_ PNCs. The calculated external quantum efficiency (EQE) is represented by discontinued lines. (**b**) One response cycle of the MPA treated photodetector at 0 V bias when illuminated under 50 Hz chopped 25 mW cm^-2^ white light.

**Figure 4 nanomaterials-10-01297-f004:**
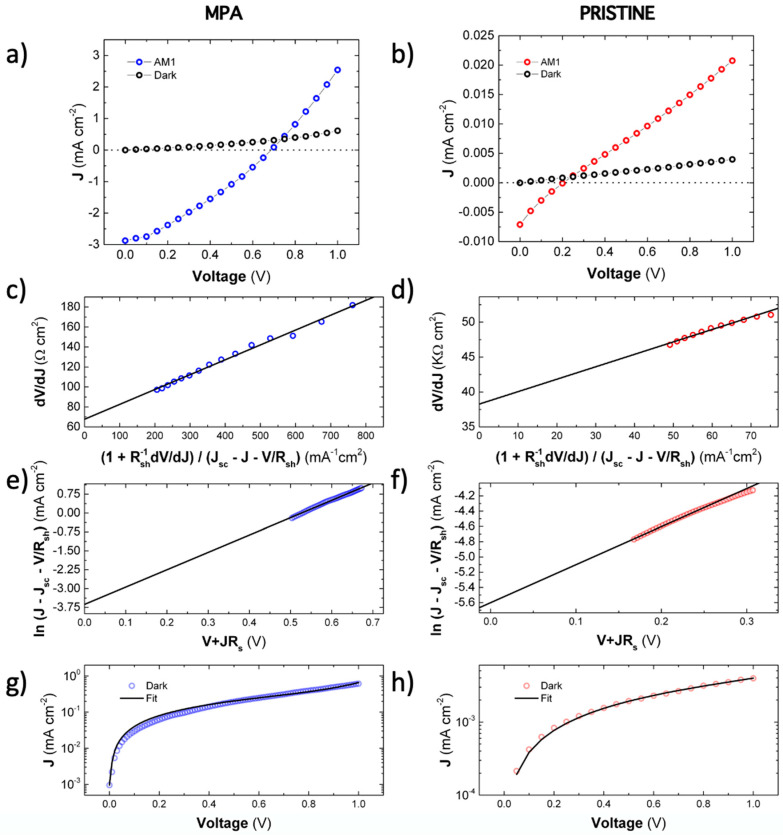
*J*-*V* curves for planar PNC photodiodes using ITO/PEDOT:PSS/CsPbBr_3_ PNCs/MoO_3_/Au architecture. The measurements are carried out in dark and under AM1.5G for MPA ligand-exchanged (**a**) and pristine (**b**) devices. Fitting curves (continuous black lines) using Equations (4)–(5) for *J*-*V* curves under AM1.5G illumination for MPA (**c**–**e**) and pristine (**d**–**f**) devices, and Equation (7) for *J*-*V* under dark conditions in MPA (**g**) and pristine (**h**) cases.

**Figure 5 nanomaterials-10-01297-f005:**
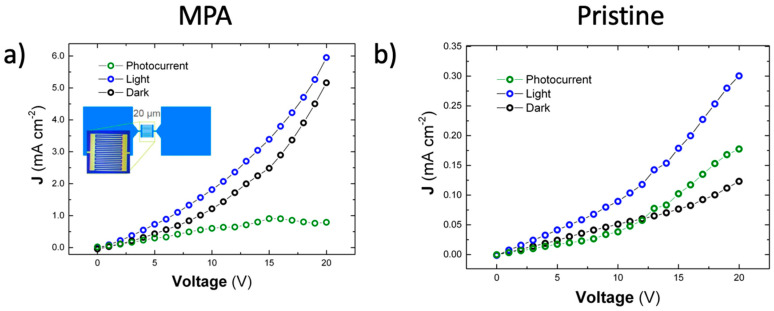
J–V characteristics of the device in dark conditions and under illumination with 25 mW cm^-2^ white light for (**a**) MPA-treated devices and (**b**) pristine devices. The photocurrent (difference between the light and the dark curves) is plotted in green color. The inset shows one test chip blueprint and optical microscopic image before the deposition of the CsPbBr_3_ PNCs.

**Figure 6 nanomaterials-10-01297-f006:**
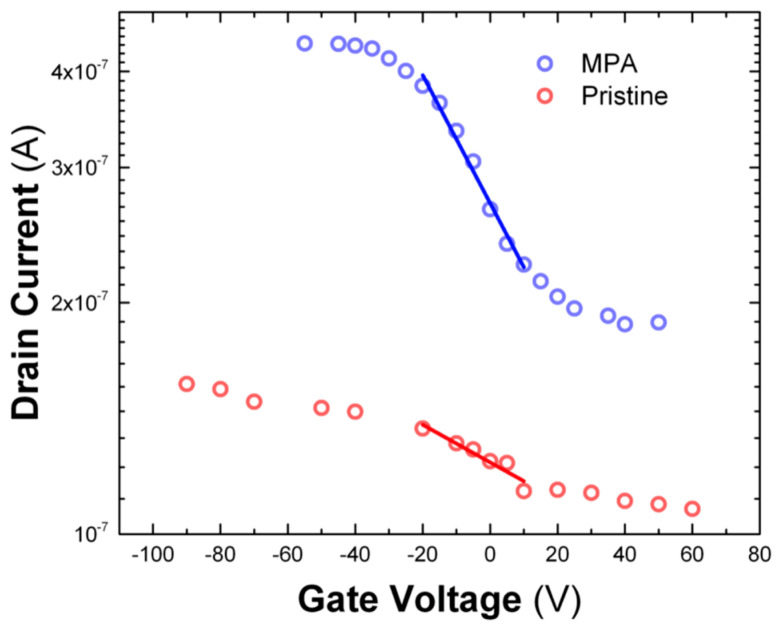
Transfer curves of processed CsPbBr_3_ PNCs-based FET devices treated with MPA (blue curve) and pristine (red curve).

**Figure 7 nanomaterials-10-01297-f007:**
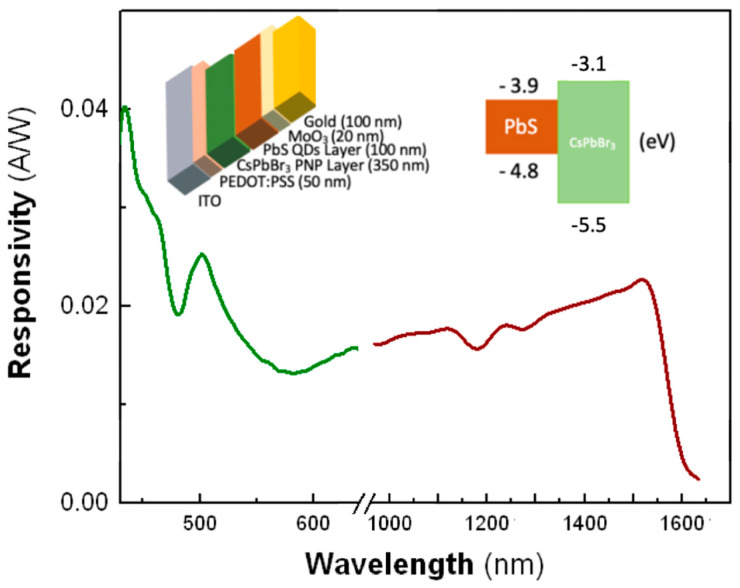
Responsivity curve of the MPA ligand-exchanged PbS quantum dots (QDs)-CsPbBr_3_ PNCs heterojunction photodiode. Device architecture is represented in the top inset. The red solid curve represents the extended responsivity range added by the PbS QDs layer while the green solid curve characterizes the responsivity part of the spectrum dominated by CsPbBr_3_ PNCs. The gap in between the two curves (800–900 nm) is due to the switching between reference photodiodes: calibrated Si and Ge detectors and different gratings for measuring in visible and near-infrared windows.

**Table 1 nanomaterials-10-01297-t001:** Average parameters of the device performances of the prepared thin film CsPbBr_3_ PNCs-based photodetectors, as obtained from the different fittings.

Sample	$R_{sh}\;(M\Omega {\mathrm{cm}}^{-2})$	$R_{s}\;(\Omega \;{\mathrm{cm}}^{-2})$	$\left(m_{1}+m_{2}\right)$	$J_{r}\;(\mathrm{mA}\;{\mathrm{cm}}^{-2})$	$J_{d}\;(\mathrm{mA}\;{\mathrm{cm}}^{-2})$	$J_{0}\;(\mathrm{mA}\;{\mathrm{cm}}^{-2})$	$\phi_{b}\;(\mathrm{eV})$
**Pristine**	4 ± 2	(3 ± 1) × 10^4^	6 ± 2	(5 ± 2) × 10^−8^	(5 ± 3) × 10^−8^	(2 ± 1) × 10^−3^	(5 ± 1) × 10^−1^
**MPA**	8 ± 3	50 ± 20	5 ± 1	(4 ± 1) × 10^−5^	0	(1.5 ± 1.0) × 10^−1^	(5 ± 1) × 10^−1^

**Table 2 nanomaterials-10-01297-t002:** Dark conductivity ($\sigma_{0}$), photoconductivity (Δσ), photoconductive sensitivity (S), and Mott–Gurney mobility (μ), extracted with Equation (8), of MSM photodetectors based on CsPbBr_3_ PNCs. The pristine and MPA-treated thin films parameters were extracted from *J*–*V* curves in Figure 5.

Sample	$\sigma_{0}\left(\mu S/cm\right)\;$	Δσ (μS/cm)	$S_{10V}=\left(J_{L}-J_{D}\right)/J_{L}$	$\mu_{SCLC}\;\left(cm^{2}/Vs\right)$
**MPA**	0.8	0.33	0.35	10^−3^
**Pristine**	0.03	0.03	0.40	10^−4^

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
