# Peer review of "Ligand-Length Modification in CsPbBr3 Perovskite Nanocrystals and Bilayers with PbS Quantum Dots for Improved Photodetection Performance"

_nanomaterials, 2020, doi:10.3390/nano10071297_

Round 1

Reviewer 1 Report

This is an experimental paper where CsPbBr3 nanocrystals were synthesized and used as the photo absorber in a photodetector device. The synthesis method used is conventional hot-injection method but the authors used a custom-made purification method to obtain high purity PNCs. Finally, ligand exchange was used to replace the long OAm ligands in the as-prepared PNC by shorter MPA ligands to improve electronic transport across the NCs.

Authors show many fold increase in the photoresponse when the film is ligand exchanged with MPA. They attribute this improvement to the shorter ligands exclusively.

Most of the discussion to explain the results uses photo physics mechanisms but actually nothing significant is developed after these complex analyses. Authors may want to revise the discussion part to discuss those mechanisms only that could prove the SRH conduction mechanism that was initially presumed. I feel there is too much speculative discussion which could be cut down. The core of the paper is actually the photo physics rather than nanomaterials.

Below are some specific comments to authors to improve the manuscript.

  1. Page 2, Line 91: define SH.
  2. Page 3, Line 113: check grammar “…..the mixture was N2 purged and …..”
  3. Fig 1b: TEM inset is too small to see the crystal size. Could authors give a higher magnification micrograph to show the crystal size is actually 8-9 nm.
  4. Page 6, Line 211: Authors mention that Fig S2 gives the film architectures but this is not correct. Fig S2 only gives the band alignment diagrams. Authors should provide schematics of the device architecture labelling all the layers, at least in cross sections, as the architectures are quite complex.
  5. Fig 3b: What is the response curve like for the pristine device? Give an example in SI for comparison.

Reviewer 2 Report

1. Line 82, “In this work, we propose MPA as a ligand exchange strategy to improve the charge carrier transport in solid thin films of CsPbBr3 PNCs photodiode detectors.”

What kinds of transport properties do the authors intend to discuss? The authors can describe the transport properties specifically.

2. Line 234, “In order to gain insight on the physical magnitudes determining charge transport and charge separation in our the Schottky-heterostructure devices, we focus on their J-V curves and the observed changes after ligand exchange.”

As the authors spent many paragraphs in describe the extraction of the diode parameters for the device, are there any insight and further physical explanation about the direct correlation between the charge transport and JV results (diode parameters) presented in the manuscript?

The device treated with MPA has the better diode parameters. In terms of the charge transport, how to interpret the results of the extracted diode parameters?

3. Line 186, “Intensity–voltage characteristics under dark and illumination conditions were measured using a Keithley's Series 2400 Source Measure Unit (SMU).”

What does “intensity” mean in this description?

4. Line 187, “The I-V curves were taken under AM1 illumination using a commercial solar simulator unit.”

Any particular reasons to employ the solar spectrum of AM1 for the measurements of the PNC photodiodes? If considering these devices as photovoltaic devices, the solar spectrum of AM1.5G is typically employed for terrestrial applications.

What is the light intensity for the measurements?

How the light intensity is calibrated?

What are the device areas?

In line 187, the authors depict that the I-V curves are measured. However, the results shown in Fig. 4 are J-V curves.

5. Line 244, “In any case, measured values of Jsc and Voc for MPA-capped CsPbBr3 PNCs devices are comparable with previously published results [37–40].”

A table for the measured results and the published results in the references is suggested to present in the manuscript, which will be beneficial to the potential readers.

6. Line 257, “Jsc is the short-circuit current and ?0 is the reverse saturation current.”

They are current densities instead of just currents.

7. Line 272, “The equivalent electrical circuit of the double heterojunction model [49] is used here to obtain the photodiode parameters of the MPA-treated photodiodes and the pristine cells…”

Since the authors employ a double junction mode to extract the parameters, is there any fact or direct evidence revealing that the device has two heterojunctions? What are these two junctions? Please name them in the manuscript. The verification of two junctions for the fabricated devices is essential.

8. The ideality factor result of “the pristine (5 ± 2) photodiodes” described in line 287 conflicts that given in Table 1.

9. Line 322, “the 321 photodiodes can still reach a good responsivity level when compared with other reported in recent literature [11,13,53].”

The responsivity of the device presented in this study is relative small as compared to the results reported in the literature, as the results shown in Table S1.

In addition, Table S1 is suggested to directly present in the manuscript instead of the supplementary, which is beneficial to the potential readers.

10. The value of mobility (10-3) for the device treated with MPA given in Table 2 is not the same as that shown in the abstract (line 18).

Reviewer 3 Report

see attached.

Round 2

Reviewer 1 Report

Authors have satisfactorily attended to all the comments and criticisms raised in the first submission. The manuscript is suitable for publication in Nanomaterials now.

Author Response

We would like to thank again the referee for the careful readthrough of our work and pertinent remarks. We have made a complete English revision of the manuscript.

Reviewer 2 Report

1. Line 169, ” This multi-electrode mask defines up to eight devices of area 4 mm2 per substrate.”

Is the area of one device 4mm2? Or, Is the area of 8 devices 4mm2? The size of a single device is suggested to describe.

2. Line 193, “…and a solar simulator based on 193 a 150 W Xe lamp (Zolix, model GORIA-X150A) and an AM1.5G air mass filter (Newport), which was 194 calibrated by means of a tabulated Si solar cell.”

2a. What does the ‘tabulated Si solar cell’ exactly mean? Please state clearly to avoid ambiguity.

2b. The Si reference cell is typically employed to calibrate the illumination intensity of the solar simulator instead of the solar spectrum. Please clarify.

2c. The illumination intensities for this study were not described.

3. Line 172, “Tandem photodevices based on double films of CsPbBr3 PNCs and PbS QDs were fabricated by using similar procedures described above.”

Line 461, “The tandem photodetector reaches a…”

Do the authors consider the tandem photovodevices or tandem photodetectors as a detector consisting of two different junctions, namely top junction and bottom junction? Are there two different junctions for each of the tandem photovodevices in this study? Please clarify.

4. The percent sign is missing for EQE resented in Figure 3(a) or the values of EQE for MPA treated sample are of no sense.

Reviewer 3 Report

The authors appear to have addressed my original concerns, and I think the paper could be published after some editing for English readability.

Author Response

(The authors gave the same response as above.)
